# DR.PO: DUAL REFERENCE AND PREFERENCE OPTIMIZATION FOR MACHINE UNLEARNING IN LARGE LANGUAGE MODEL

## ABSTRACT

Most high-performing LLM unlearning methods use preference learning but have a critical flaw: insufficient or singular positive preferences make post-unlearning models generate meaningless, inconsistent, or single-category outputs. These differ in probability distribution from "fully unlearned" models, causing suboptimal unlearning quality and privacy risks of unlearned data. We propose DR.PO (Dual Reference and Preference Optimization). It adopts answers with incorrect facts and answers indicating information deficiency as positive preferences, with distinct reference models for each. Instead of random sampling the full dataset for the retain set (as in existing methods), we match each forget sample to a corresponding retain sample via similarity scores—reducing repeated sampling of the retain set. Experiments show our method not only achieves better unlearning quality and better privacy protection but also effectively preserves the model's original capabilities.

## 1 INTRODUCTION

Since deep learning models require the use of massive amounts of data during the training process, concerns have been raised by relevant personnel regarding data privacy, copyright, and security (Mantelero, 2013; de la Torre, 2018; Wang et al., 2024). Consequently, machine unlearning for LLMs has attracted significant attention from researchers. In previous studies, researchers have explored techniques such as gradient ascent, token-level perturbation, and preference learning to implement machine unlearning in LLMs, achieving certain results. Among these, preference learning methods often demonstrate superior performance.

However, due to the absence or singularity of positive preferences in existing preference learning methods, there exists a discrepancy between the output distribution of the unlearned model when exposed to the forgotten data and that of a model that never learned the forgotten data in real scenarios. This makes it challenging to strike an effective balance between unlearning quality and privacy protection of the forgotten data. To address this, we propose **Dual Reference and Preference Optimization (DR.PO)**, a preference learning method that employs dual positive preferences paired with corresponding reference models. By leveraging dual positive preferences to guide the model's unlearning process, DR.PO enables the model to produce an output distribution closer to that of exact unlearning when encountering the forget data.

Additionally, when examining existing methods, we observed that to maintain model performance, these methods typically incorporate randomly sampled samples from the retention set in each training epoch. We argue that this process is problematic in terms of computational resource consumption and stability, and may amplify the impact of certain retention set samples that are less affected by the unlearning process. Therefore, we instead use a fixed set of top-k retention samples with the highest similarity to the forgotten data in the training process, and control the influence of these retention samples based on their similarity scores.

In summary, our main contributions are as follows:

- **Dual Positive Preference:** We propose a dual positive preference forgetting method. Through the synergy of dual positive preferences, answers indicating missing information, and answers with

incorrect facts, the model can accommodate both preferences simultaneously in its outputs, thus better protecting the privacy of forgotten data.

- **Dual Reference Model:** We propose using the pre-finetuning model as the reference model for the positive preference of missing information. Experiments show that this better-aligned selection of the reference model can better exert the effect of preference guidance.

- **Similarity-scores Retain:** We propose to select the retained data participating in unlearning based on vector similarity, and use vector similarity to restrict their degree of influence. Experiments show that this method can achieve performance close to that of the original random sampling and is more stable.

## 2 RELATED WORK AND PRELIMINARIES

### 2.1 HOW CAN WE STATE LLM UNLEARNING?

Given an LLM $\pi_\theta$ with parameters $\theta$ and a dataset $D = \{(x_i, y_i)\}_{i=1}^n$, where $(x_i, y_i)$ means an question-answer pair in $D$, we denote the forget set as $D_f \subset D$, the retain set as $D_r = D - D_f$, the LLM that has been trained on $D$ as $\pi_{full}$ and the pretrained LLM has not been trained on $D$ as $\pi_{base}$. The goal of LLM Unlearning is to make the $\pi_{full}$ "forget" the specific information contained in $D_f$ - i.e., to transform $\pi_{full}$ into $\pi_{unl}$ such that the performance of $\pi_{unl}$ is approximately consistent with those of $\pi_{ret}$ only trained on $D_r$. Specially, $\pi_{unl}$ should not retain dependencies on $D_f$ and ensure the unlearning process only eliminates the influence of $D_f$ without distorting the model's utility derived from $D_r$ and other datasets.

### 2.2 WHAT ARE THE EXISTING LLM UNLEARNING METHODS?

Due to the massive parameter size of deep learning models, the method of achieving exact unlearning through retraining with a retain set faces the challenge of high computational costs in practical scenarios. Consequently, researchers tend to focus on developing a series of approximate unlearning methods, which fine-tune model parameters using a forget set. Such approaches have already yielded beneficial results in fields including federated learning (Halimi et al., 2022; Dhasade et al., 2023; Gu et al., 2024; Jin et al., 2023), lifelong learning (Du et al., 2019; Parisi et al., 2019), image generation (Li et al., 2024a; Zhang et al., 2024c; Gandikota et al., 2023; Zhang et al., 2024b), graph neural networks (Li et al., 2024c; Cheng et al., 2023; Tan et al., 2024), and recommendation systems (Sinha et al., 2025; Li et al., 2023).

In complex LLM Unlearning tasks, researchers have proposed a series of benchmark tests. These include WMDP, which focuses on the unlearning of hazardous knowledge (Li et al., 2024b); TOFU, which centers on personal privacy information (Maini et al., 2024); and MUSE, which addresses the privacy/copyright issues of data owners (Shi et al., 2025).

Methodologically, some researchers have attempted Gradient Ascent (Maini et al., 2024) and its derived methods, namely Gradient Difference (Liu et al., 2022) and WGA (Wang et al., 2025), based on the idea of Gradient Ascent; some have explored token-level unlearning methods, such as UNDIAL (Dong et al., 2025), SatImp (Yang et al., 2025), and PDU (Entesari et al., 2025); others have proposed RMU (Li et al., 2024b), which perturbs model parameters related to the data to be unlearned. Although the aforementioned methods that directly perform unlearning on model parameters have shown certain effects, their performance in terms of unlearning efficacy, model performance, and privacy protection of the unlearned data is unsatisfactory, leading to inconsistent or incoherent responses generated by the model.

Therefore, some other researchers have started from preference learning and designed a series of methods based on DPO (Rafailov et al., 2023). ($\beta$ in the following equations is the regularization strength)

**NPO** (Zhang et al., 2024a) retains only the negative preference component and guides the model to avoid generating correct responses:

$$L_{NPO} = -\frac{2}{\beta} log\sigma(-\beta log \frac{\pi_\theta(y_f|x_f)}{\pi_{full}(y_f|x_f)}) \tag{1}$$

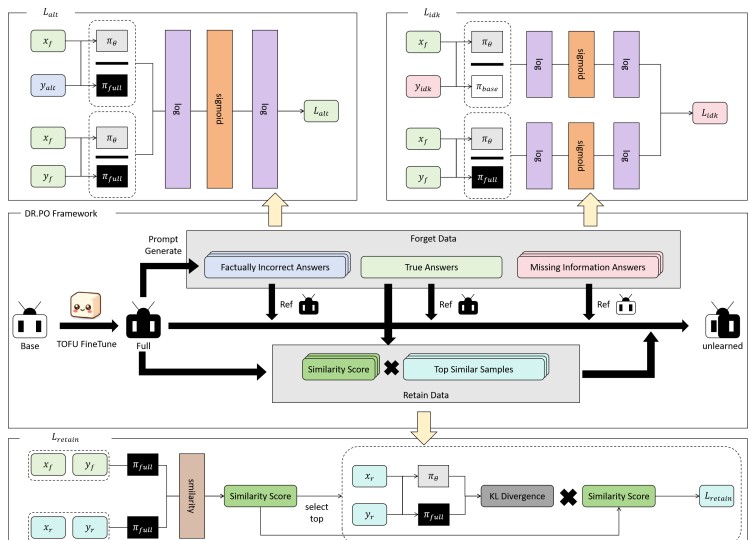

Figure 1: The framework of DR.PO.

**SimNPO** (Fan et al., 2024) is improved based on NPO: it removes the reliance on reference models and sets up a reward margin control to avoid uneven allocation of optimization resources($\gamma$ is the reward margin parameter):

$$L_{SimNPO} = \mathbb{E}[-\frac{2}{\beta}log\sigma(-\frac{\beta}{|y_f|}log\pi_\theta(y_f|x_f - \gamma)]$$ (2)

**IdkPO** (Maini et al., 2024) uses answers indicating missing information as positive preference responses, guiding the model to be more inclined to indicate missing information when dealing with unlearned data($y_{idk}$ is a response randomly selected from a number of pre-prepared answers indicating missing information):

$$L_{IdkPO} = -\frac{2}{\beta}log\sigma(\beta log\frac{\pi_\theta(y_{idk}|x_f)}{\pi_{full}(y_{idk}|x_f)} - \beta log\frac{\pi_\theta(y_f|x_f)}{\pi_{full}(y_f|x_f)})$$ (3)

**AltPO** (Mekala et al., 2025) uses prompts to require the model to generate a set of answers with incorrect facts based on the correct answers to forget data. It takes these answers with incorrect facts as positive preference responses and guides the model to provide factually incorrect answers($y_{alt}$ is the generated factually incorrect answers):

$$L_{AltPO} = \mathbb{E}_{y_{alt}}[-\frac{2}{\beta}log\sigma(\beta log\frac{\pi_\theta(y_{alt}|x_f)}{\pi_{full}(y_{alt}|x_f)} - \beta log\frac{\pi_\theta(y_f|x_f)}{\pi_{full}(y_f|x_f)})]$$ (4)

## 3 METHODS

Figure 1 shows the overview of our method.

### 3.1 DR.PO: DUAL REFERENCE AND PREFERENCE OPTIMIZATION

Although Preference Optimization (PO)-type methods have achieved better unlearning performance, due to the way positive preferences are selected, they often perform poorly in terms of the privacy of forget data. Specifically, IdkPO only selects missing information answers as positive preferences, and AltPO only selects factually incorrect responses as positive preferences. All these issues result in a tendency toward a single response pattern.

For the reasons as mentioned above, and considering the fact that LLMs randomly generate either answers indicating missing information or answers with incorrect fact when addressing unknown

questions—with the former typically being what humans expect and require the model to output during fine-tuning (He et al., 2025), and the latter arising from the hallucination phenomenon of LLMs—we propose a method that can simultaneously integrate the positive preferences of IdkPO and AltPO, namely **DR.PO: Dual Reference and Preference Optimization**.

Building on the integration of IdkPO and AltPO losses (Maini et al., 2024; Mekala et al., 2025), DR.PO modifies the loss function of IdkPO. Since the effectiveness of Preference Optimization (PO)-type methods depends to some extent on the choice of reference model, and PO is conceptually based on responses that the model itself could generate with a certain probability, it is clearly more appropriate to use a pre-trained model $\pi_{base}$ that has not been exposed to the unlearned data as the reference model for responses indicating missing information. Meanwhile, for correct responses, the model $\pi_{full}$ is still used as the reference model. Furthermore, due to the use of different reference models for positive and negative preferences, the original loss form cannot be mathematically retained; instead, we adopt two single-preference losses as substitutes.

$$L_{idk} = \mathbb{E}_{y_{idk}}[-\frac{2}{\beta}(log\sigma(\beta log\frac{\pi_\theta(y_{idk}|x_f)}{\pi_{base}(y_{idk}|x_f)}) + log\sigma(-\beta log\frac{\pi_\theta(y_f|x_f)}{\pi_{full}(y_f|x_f)}))] \quad (5)$$

$$L_{alt} = \mathbb{E}_{y_{alt}}[-\frac{2}{\beta}log\sigma(\beta log\frac{\pi_\theta(y_{alt}|x_f)}{\pi_{full}(y_{alt}|x_f)} - \beta log\frac{\pi_\theta(y_f|x_f)}{\pi_{full}(y_f|x_f)})] \quad (6)$$

$$L_{DR.PO} = L_{idk} + L_{alt} \quad (7)$$

where $\beta$ is the regularization strength, as in other PO methods.

## 3.2 RETAIN DATA PROCESSING BASED ON KL AND VECTOR SIMILARITY

Most LLM Unlearning methods involve fine-tuning with part of the retained dataset. Typically, random sampling is performed on the complete retained dataset, and a subset of the retained dataset is used for training. This approach can be traced back to Gradient Difference (Liu et al., 2022), which adds the log-likelihood loss on a random subset of the retained dataset to the loss of the unlearned dataset—a practice that has been adopted consistently by subsequent methods (Zhang et al., 2024a; Maini et al., 2024; Mekala et al., 2025).

Although this approach to using the retained dataset has been proven effective on benchmarks, we still have concerns: random sampling means that the coverage is completely random. While it may achieve effective coverage for small datasets, for large datasets, it will likely require sampling more samples to maintain model performance, thereby increasing computational costs. Based on our intuition, we propose selecting only retained dataset samples with high similarity to the unlearned samples for training. We use vector similarity as a coefficient and utilize these retained dataset samples in the form of KL divergence (Maini et al., 2024) to constrain the similarity of responses on the retained dataset between the model undergoing unlearning and the non-unlearned model, which we have proven effective.

$$L_{retain} = \sum_{i=1}^{k} sim(\pi_{full}(x_r^{(i)}, y_r^{(i)}), \pi_{full}(x_f, y_f))KL(\pi_{full}(y_r^{(i)}|x_r^{(i)})||\pi_\theta(y_r^{(i)}|x_r^{(i)})) \quad (8)$$

$$L_{DR.PO-with-retain} = L_{idk} + L_{alt} + L_{retain} \quad (9)$$

where $k$ represents the number of top-k retained data set samples selected based on the highest similarity to the unlearned samples. $sim$ computes the vector similarity between two distributions; we use the vector of the last token from the output of the model's final hidden layer as the feature vector for the input question-answer pair. $KL$ measures the divergence between the output distributions of the two models.

## 4 EXPERIMENTS

### 4.1 BENCHMARK

Proposed in 2024, TOFU is currently the most commonly used LLM Unlearning benchmark, on which most existing methods have been evaluated. It mainly includes information on 200 fictional

| Data Type | TOFU 1% | TOFU 5% | TOFU%10 |
|---|---|---|---|
| Forget data | 40(2 authors) | 200(10 authors) | 400(20 authors) |
| Retain data | 3960(198 authors) | 3800(190 authors) | 3600(180 authors) |
| All Finetune data | 4000(200 authors) | | |

Table 1: The Information of TOFU.

authors, sets three difficulty levels (1%, 5%, and 10% unlearning)(see in Table 1), and provides a variety of model evaluations. We selected TOFU as the benchmark for our experiments and chose Llama3.2-1B as the experimental model based on computational resources. (Maini et al., 2024)

Meanwhile, we used OpenUnlearning as our evaluation framework and the implementation framework for baselines. Proposed in 2025, OpenUnlearning is an LLM Unlearning framework that includes complete evaluation functionality for the TOFU Benchmark and provides implementations of a series of baseline methods. (Dorna et al., 2025)

## 4.2 METRICS

Based on the four aspects that need to be considered in LLM Unlearning - accuracy of unlearning, knowledge preservation, privacy protection training, and fluency in dialogue generation - we selected four metrics, namely Forget Quality, Model Utility, Privacy Leakage, and Forget Gibberish - for evaluation (Maini et al., 2024; Shi et al., 2025; Dorna et al., 2025).

**Forget Quality(FQ):** The metric proposed by TOFU for measuring forget quality compares the truth ratio distributions of two models through the Kolmogorov-Smirnov Test (KS-Test), and quantifies forget quality using the p-value of the test: a higher p-value indicates a smaller difference between the distributions of the two models, thus a better forgetting effect. (Maini et al., 2024)

**Model Utility(MU):** The metric proposed by TOFU to measure the extent to which a model retains performance on non-unlearned data aggregates three indicators—probability, ROUGE-L, and truth ratio—across three datasets (Retain Set, Real Authors, and World Facts) using the harmonic mean. (Maini et al., 2024)

**Privacy Leakage(Priv.):** MUSE proposes that although most existing unlearning algorithms can reduce verbatim or knowledge memorization, they generally suffer from severe privacy leakage—either "over-unlearning" leading to abnormal behaviors or "under-unlearning" retaining training traces. Therefore, this metric is designed to measure the degree of privacy leakage: a value greater than 0 indicates "over-unlearning," while a value less than 0 indicates "under-unlearning." (Shi et al., 2025; Hayes et al., 2024)

**Forget Gibberish(Gibb.):** OpenUnlearning proposed this metric to measure whether a model exhibits the phenomenon of "gibberish" on the forget set after unlearning. OpenUnlearning uses the "autonlp-Gibberish-Detector-492513457" model to evaluate outputs, where a lower score indicates a higher likelihood that the output text is "gibberish". (Dorna et al., 2025; Jindal, 2021)

## 4.3 IMPLEMENTATION DETAILS

All our experiments were conducted on a single NVIDIA A800 GPU. For $y_{alt}$ in $L_{alt}$, we used the generation script of AltPO to generate factually incorrect answers in the default recommended quantity of AltPO, i.e., five answers. Meanwhile, for $y_{idk}$ in $L_{idk}$, we selected five answers from the 100 optional answers provided by IdkPO in each epoch. For the baselines, we configured the experimental settings by referring to the recommended parameters preset in the corresponding papers and open-unlearning. Details can be found in the appendix.

## 4.4 RESULTS

### 4.4.1 COMPARATIVE EXPERIMENT

Table 2 presents the experimental results of our method and other existing methods on the TOFU Benchmark under three settings: 1%, 5%, and 10%. From the results, it can be observed that methods of the PO type are more likely to achieve superior performance compared to other methods. In particular, AlterPO has basically reached the SOTA level, while other methods (Grad Ascent,

Table 2: Performance of various unlearning methods of TOFU 1%, 5% and 10% on Llama3.2-1B. We use (↑) to indicate that a higher metric value is better, and (→ 0) to indicate that a metric value closer to 0 is better. In the "FQ" and "Priv." columns, the optimal results are highlighted in bold, and the suboptimal results are underlined; in the "MU" column, results that are close to those of "Full" and "Retain" are highlighted in bold, indicating no impairment caused by unlearning.

| Method | TOFU 1% | | | | TOFU 5% | | | | TOFU 10% | | | |
| | FQ (↑) | MU (↑) | Priv. (→ 0) | Gibb. (↑) | FQ (↑) | MU (↑) | Priv. (→ 0) | Gibb. (↑) | FQ (↑) | MU (↑) | Priv. (→ 0) | Gibb. (↑) |
|---|---|---|---|---|---|---|---|---|---|---|---|---|
| Full | 6.76e-03 | 0.5992 | -100.0000 | 0.8944 | 1.43e-12 | 0.5992 | -99.9922 | 0.8584 | 3.91e-22 | 0.5991 | -99.4574 | 0.8606 |
| Retain | 1.00e+00 | 0.5986 | 0.0000 | 0.8739 | 1.00e+00 | 0.5991 | 0.0000 | 0.9045 | 1.00e+00 | 0.5911 | 0.0000 | 0.9043 |
| Grad Ascent | 2.86e-02 | **0.5903** | -59.5041 | **0.8979** | 1.94e-119 | 0.0000 | -31.6578 | 0.1030 | 1.06e-239 | 0.0000 | -17.8960 | 0.3702 |
| Grad Difference (Liu et al., 2022) | 2.86e-02 | 0.5206 | -62.1015 | 0.8183 | 8.05e-07 | 0.4497 | -13.1744 | 0.7453 | 1.60e-12 | 0.4365 | 20.2932 | 0.4159 |
| WGA (Wang et al., 2025) | 2.86e-02 | **0.5993** | -86.8123 | **0.8847** | 2.96e-05 | **0.5952** | -42.8227 | **0.8475** | 4.46e-06 | **0.5984** | **3.2658** | 0.7086 |
| UNDIAL (Dong et al., 2025) | 1.43e-02 | **0.6097** | -83.8253 | 0.6561 | 6.57e-12 | **0.6090** | -96.6692 | 0.7469 | 2.21e-19 | 0.5973 | -97.0055 | 0.6709 |
| SatImp (Yang et al., 2025) | 1.43e-02 | **0.6023** | -98.7485 | **0.9184** | 1.21e-10 | **0.5965** | -97.6357 | **0.8977** | 1.18e-17 | **0.5996** | -97.1979 | **0.8889** |
| RMU (Li et al., 2024b) | 5.41e-02 | 0.5595 | -49.4687 | 0.7212 | 8.78e-02 | 0.5743 | 21.8240 | 0.5488 | 7.75e-04 | 0.5831 | 59.0735 | 0.0386 |
| PDU (Entesari et al., 2025) | 6.76e-03 | **0.6036** | -100.0000 | 0.8744 | 5.951e-11 | **0.5974** | -98.5296 | 0.8095 | 1.371e-07 | 0.5837 | -40.1460 | 0.5656 |
| NPO (Zhang et al., 2024a) | 1.65e-01 | **0.5977** | -48.4061 | **0.9202** | 4.31e-03 | 0.5939 | -68.7696 | **0.9016** | 3.09e-06 | **0.5957** | -70.5473 | **0.8895** |
| SimNPO (Fan et al., 2024) | 2.86e-02 | **0.5982** | -83.2349 | **0.9028** | 2.38e-06 | 0.5918 | -78.8347 | **0.8834** | 3.65e-11 | 0.5954 | -74.8132 | **0.8711** |
| IdkPO (Maini et al., 2024) | 5.41e-02 | 0.5837 | -72.7273 | **0.9016** | 1.39e-06 | 0.5901 | -81.1598 | **0.9082** | 7.83e-12 | 0.5905 | -76.2882 | **0.9030** |
| AltPO (Mekala et al., 2025) | 9.19e-01 | 0.5952 | -54.3329 | 0.8580 | **9.24e-01** | 0.5891 | -30.0012 | 0.8460 | 5.23e-01 | 0.5844 | -9.6435 | 0.8528 |
| DR.PO(Ours) | **9.90e-01** | 0.5954 | **0.9445** | 0.9315 | 8.66e-01 | 0.5960 | **-12.0491** | 0.8995 | **8.64e-01** | 0.6020 | 7.7646 | 0.8933 |

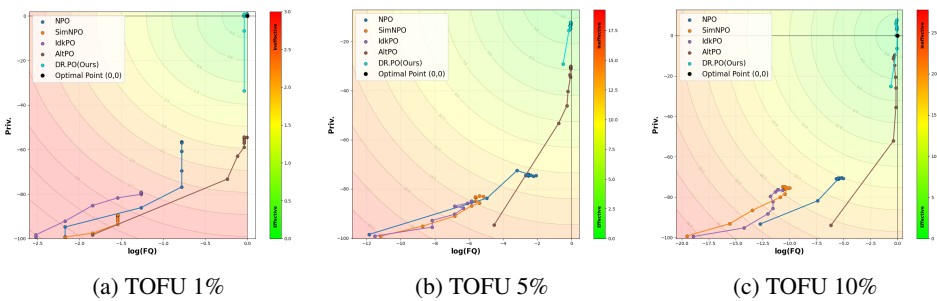

(a) TOFU 1%      (b) TOFU 5%      (c) TOFU 10%

Figure 2: The corresponding relationship between Forget Quality and Privacy Leakage during the unlearning process of TOFU 1%, TOFU 5%, and TOFU 10% on Llama3.2-1B of various PO-type unlearning methods.

Grad Difference, WGA, RMU, UNDAIL) tend to exhibit performance degradation relative to the original model.

Across all three scales, our method achieves a better balance between Forget Quality and Privacy Leakage. Specifically, under the 1% and 10% settings, our method reaches SOTA in both Forget Quality and Privacy Leakage, demonstrating better forgetting effectiveness and privacy compared to existing methods. Under the 5% setting, compared with AlterPO (which achieves SOTA in Forget Quality), our method obtains significantly superior Privacy Leakage at the cost of slightly lower Forget Quality—indicating that our method offers better privacy with comparable forgetting quality.

Meanwhile, compared to other methods, our method also maintains excellent performance in the two metrics of Model Utility and Forget Gibberish, which shows that our method can effectively avoid performance damage to the model caused by unlearning.

Figure 2 shows the combined variation of Forget Quality and Privacy Leakage during the unlearning process of PO-type methods. It can be seen that as Forget Quality increases, Privacy Leakage will also increase and eventually tend to be stable. The curves show that compared with other methods, our method often has better Privacy Leakage at the same Forget Quality, which indicates that our method has a better effect on the privacy protection of forgotten data than other methods.

In order to examine the impact of double positive feedback on the final answers generated by the model, we additionally used the unlearned models obtained using various PO-type unlearning methods to generate responses for questions related to the forget data. We classified the generated answers into categories using the Deepseek API: Irrelevant Answer, Missing Information Answer, or Factually Incorrect Answer. For each question related to the forget data, we instructed the model to

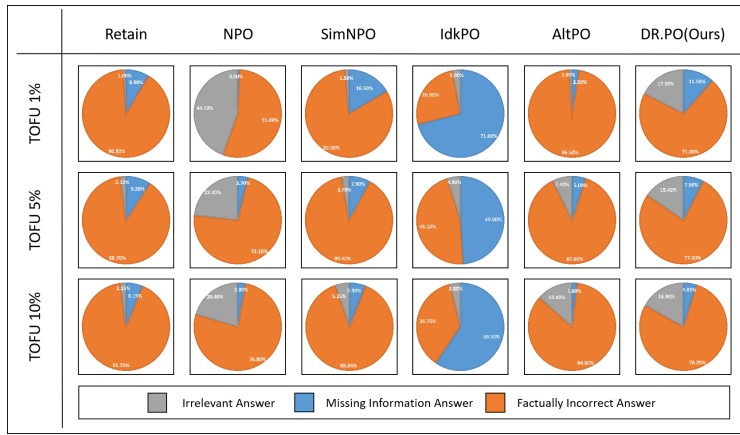

Figure 3: Regarding data forgetting questions, the proportion of various types of answers generated by different PO-type unlearning methods

generate 5 different responses and classify these answers using the Deepseek API (Liu et al., 2024; Guo et al., 2025). Figure 3 presents our experimental results. It can be observed that, compared to IdkPO and AltPO, our method is closer to the retain model in terms of the distribution of Missing Information Answers. However, our method produces more Irrelevant Answers, though it still outperforms NPO, which does not use positive preference. Meanwhile, the distribution of the three types of answers generated by our method is more stable compared to other methods—it does not exhibit significant changes due to fluctuations in the forgetting percentage, which is more consistent with the behavior of the retain model. This demonstrates that our method is indeed closer to the performance of the retain model when answering questions, and thus better protects the privacy of the forget data.

### 4.4.2 ABLATION EXPERIMENT ON FORGET LOSS

To verify the effectiveness of the design of the forgetting loss in our method, we conducted ablation experiments on the forgetting loss component of our method. Our method mainly features the following aspects: (1) Adoption of a dual positive preference approach; (2) For answers indicating missing information, the model $\pi_{base}$ without fine-tuning on $D$ is used as the reference model; (3) For answers indicating missing information, a dual one-way preference superposition format is employed due to differences in reference models; (4) For answers indicating missing information, we adopt the same method of calculating expectations over multiple samples as that used for factually incorrect responses.

**Preference Ablation(no_idk, no_alt):** To verify Feature (1), we ablated $L_{idk}$ and $L_{alt}$ from the forgetting loss respectively and observed the effects.

**Reference Model Replacement(idk_ref_full):** To verify Feature (2), we replace $L_{idk}$ with a form that, like $L_{alt}$, has dual preference outputs and both take $\pi_{full}$ as the reference model. In this case, $L_{idk}$ changes to:

$$L_{idk} = \mathbb{E}_{y_{idk}}[-\frac{2}{\beta}log\sigma(\beta log\frac{\pi_\theta(y_{idk}|x_f)}{\pi_{full}(y_{idk}|x_f)} - \beta log\frac{\pi_\theta(y_f|x_f)}{\pi_{full}(y_f|x_f)})] \tag{10}$$

**Don't Split Idk(idk_no_split):** To verify Feature (3), we retain the original form of the DPO loss and only modify the reference model in the positive preference part to $\pi_{base}$. At this point, $L_{idk}$ is changed to:

$$L_{idk} = \mathbb{E}_{y_{idk}}[-\frac{2}{\beta}log\sigma(\beta log\frac{\pi_\theta(y_{idk}|x_f)}{\pi_{base}(y_{idk}|x_f)} - \beta log\frac{\pi_\theta(y_f|x_f)}{\pi_{full}(y_f|x_f)})] \tag{11}$$

**Use Only One Idk(idk_single):** To verify Feature (4), we adopt the same approach as IdkPO: randomly selecting a single answer indicating missing information and applying it to $L_{idk}$, replacing the original form of sampling multiple answers and calculating the expectation.

Table 3: Performance of various forget loss ablation methods of TOFU 1% and TOFU 5% on Llama3.2-1B.

| Method | TOFU 1% | | | | TOFU 5% | | | | TOFU 10% | | | |
|---|---|---|---|---|---|---|---|---|---|---|---|---|
| | FQ (↑) | MU (↑) | Priv. (→ 0) | Gibb. (↑) | FQ (↑) | MU (↑) | Priv. (→ 0) | Gibb. (↑) | FQ (↑) | MU (↑) | Priv. (→ 0) | Gibb. (↑) |
| Full | 6.76e-03 | 0.5992 | -100.0000 | 0.8944 | 1.43e-12 | 0.5992 | -99.9922 | 0.8584 | 3.91e-22 | 0.5991 | -99.4574 | 0.8606 |
| Retain | 1.00e+00 | 0.5986 | 0.0000 | 0.8739 | 1.00e+00 | 0.5991 | 0.0000 | 0.9045 | 1.00e+00 | 0.5911 | 0.0000 | 0.9043 |
| DR.PO | 9.90e-01 | 0.5954 | 0.9445 | 0.9315 | 8.66e-01 | 0.5960 | -12.0491 | 0.8995 | 8.64e-01 | 0.6020 | 7.7646 | 0.8933 |
| DR.PO(no_idk) | 9.19e-01 | 0.5979 | -60.2125 | 0.8646 | 1.78e-01 | 0.5982 | -59.4064 | 0.8957 | 3.22e-01 | 0.5978 | -33.6984 | 0.8840 |
| DR.PO(no_alt) | 7.66e-01 | 0.5973 | 15.2302 | 0.9239 | 4.31e-03 | 0.5968 | -17.5306 | 0.8848 | 1.78e-04 | 0.5941 | 20.2476 | 0.8799 |
| DR.PO(idk_ref_full) | 9.19e-01 | 0.5960 | -44.9823 | 0.8748 | 1.63e-02 | 0.5992 | -73.5492 | 0.9139 | 1.55e-01 | 0.6004 | -49.1233 | 0.8718 |
| DR.PO(idk_no_split) | 9.19e-01 | 0.5953 | -48.4061 | 0.8982 | 6.80e-02 | 0.5981 | -69.5067 | 0.9014 | 7.83e-02 | 0.6009 | -45.8525 | 0.8678 |
| DR.PO(idk_single) | 9.90e-01 | 0.5944 | 30.5785 | 0.9053 | 8.66e-01 | 0.5984 | -9.1672 | 0.9007 | 7.00e-01 | 0.5955 | 3.0603 | 0.8814 |

Table 3 presents the experimental results of our ablation study on the forgetting set loss:

**Two positive preferences collaborate:** When either $L_{idk}$ or $L_{alt}$ is ablated, there is a significant decline in both the forgetting quality and the privacy protection performance of the forgetting set. This fully demonstrates the effectiveness and necessity of simultaneously using factually incorrect responses and responses indicating information deficiency as positive preferences. Specifically, the ablation of $L_{idk}$ is more prominently manifested as a drop in privacy protection performance, while the ablation of $L_{alt}$ is more notably reflected in a decrease in forgetting performance. This indicates that taking factually incorrect responses as positive preferences mainly dominates the quality of forgetting, whereas using responses indicating information deficiency as positive preferences primarily helps achieve the privacy protection of the forgetting set.

**It is necessary to match the reference model with the preference:** When $L_{idk}$ is modified into a form where both positive and negative preferences take $\pi_{full}$ as the reference model, there is also a significant decline in both the forgetting quality and the privacy protection performance of the forgetting set. This indicates that using an inappropriate reference model for the corresponding preferences can also affect the performance.

**When the reference models differ, the dual one-way preference outperforms the original DPO:** When $L_{idk}$ is modified into the form of DPO loss and only the reference model in the positive preference part is adjusted, both the forgetting quality and the forgetting set privacy protection performance of DR.PO also deteriorate. It can be seen that when the reference models differ, it is not a good choice to adhere to the original DPO form rigidly; instead, using a dual one-way preference for simulation can well replace the original DPO form.

**The positive preference should adopt the form of multiple expectations:** When a single sample is randomly selected for the calculation of $L_{idk}$, the forgetting quality is mostly comparable to that when multiple samples are randomly selected. Specifically, on TOFU 5% and TOFU 10%, there is even a slight improvement in the privacy protection performance of the forgetting set. However, on TOFU 1%, a distinct positive impulse signal emerges in the indicator for the forgetting set's privacy protection performance. This indicates that when the data volume of the forgetting set is large, the number of sampled items has no significant impact on performance, as a sufficient number of positive preference samples can be learned. In contrast, when the data volume of the forgetting set is small, over-unlearning is prone to occur due to the limited number of positive preference samples available for learning.

### 4.4.3 ABLATION EXPERIMENT ON RETAIN LOSS

Our method also incorporates minor designs for the retention loss. To verify this, we conducted ablation experiments specifically on the retention loss, primarily to validate the effectiveness of two modifications: using similarity scores as coefficients and adopting KL divergence to replace the traditional NLL (Negative Log Likelihood) as the loss function, as well as the equivalence of using the top-10 most similar retention set samples instead of randomly sampling retention set samples in each iteration.

**no_sim** indicates that similarity scores are not used as coefficients, meaning the coefficient for the loss of all retention set samples is 1 when included in the calculation; **nll** denotes the use of traditional NLL (Negative Log Likelihood) to compute the loss of retention set samples; **random** refers

Table 4: Performance of various retain loss ablation methods of TOFU 1% and TOFU 5% on Llama3.2-1B.

| Method | TOFU 1% | | | | TOFU 5% | | | | TOFU 10% | | | |
|---|---|---|---|---|---|---|---|---|---|---|---|---|
| | FQ (↑) | MU (↑) | Priv. (→ 0) | Gibb. (↑) | FQ (↑) | MU (↑) | Priv. (→ 0) | Gibb. (↑) | FQ (↑) | MU (↑) | Priv. (→ 0) | Gibb. (↑) |
| Full | 6.76e-03 | 0.5992 | -100.0000 | 0.8944 | 1.43e-12 | 0.5992 | -99.9922 | 0.8584 | 3.91e-22 | 0.5991 | -99.4574 | 0.8606 |
| Retain | 1.00e+00 | 0.5986 | 0.0000 | 0.8739 | 1.00e+00 | 0.5991 | 0.0000 | 0.9045 | 1.00e+00 | 0.5911 | 0.0000 | 0.9043 |
| DR.PO | 9.90e-01 | 0.5954 | 0.9445 | 0.9315 | 8.66e-01 | 0.5960 | -12.0491 | 0.8995 | 8.64e-01 | 0.6020 | 7.7646 | 0.8933 |
| DR.PO(no_sim) | 9.19e-01 | 0.5946 | 32.9398 | 0.8816 | 7.93e-01 | 0.5961 | -13.2371 | 0.9000 | 3.67e-01 | 0.6005 | 16.6630 | 0.8737 |
| DR.PO(nll) | 2.66e-01 | 0.5382 | 24.6753 | 0.8729 | 2.83e-04 | 0.4934 | 29.5483 | 0.8462 | 4.36e-09 | 0.4725 | 28.0162 | 0.8754 |
| DR.PO(random) | 9.90e-01 | 0.5952 | -7.2019 | 0.9009 | 5.45e-01 | 0.5981 | -25.6038 | 0.8772 | 7.00e-01 | 0.5999 | 2.0176 | 0.8786 |
| DR.PO(retain01) | 7.66e-02 | 0.5763 | 6.7296 | 0.9332 | 2.97e-02 | 0.5844 | -56.6382 | 0.8933 | 2.29e-03 | 0.5863 | -52.6837 | 0.8806 |
| DR.PO(retain05) | 9.19e-01 | 0.5918 | 0.4723 | 0.9285 | 3.94e-01 | 0.5908 | -26.9683 | 0.8970 | 3.67e-01 | 0.5952 | 3.4035 | 0.8888 |
| DR.PO(retain15) | 9.19e-01 | 0.5958 | 4.2503 | 0.8770 | 7.93e-01 | 0.5945 | -13.2411 | 0.8945 | 1.55e-01 | 0.5985 | 20.3003 | 0.8838 |

to the practice of sampling samples directly from the complete retention set in each epoch for retention loss calculation, rather than using the pre-prepared top-10 samples with the highest similarity scores to the forgetting set samples as the samples for retention loss calculation. Regarding the number of samples selected from the retain set, in addition to the default of 10 samples, we also attempted scenarios with 1, 5, and 15 samples, denoted as **retain01**, **retain05**, and **retain15**.

Table 4 presents our ablation results on $L_{retain}$. When replacing the KL divergence loss with the conventional NLL loss, we observed a significant decline in Model Utility. This indicates that when using a small amount of retain data highly correlated with the forget data for training, directly applying KL divergence constraints yields better results in learning the corresponding probability distribution compared to using NLL. When removing the similarity coefficient, Forgetting Quality deteriorated. This phenomenon can be attributed to the fact that without the correlation coefficient, all highly correlated retain data participated in training with equal intensity, thereby weakening the forgetting effect. When replacing direct sampling of the top-most similar samples with random sampling, our method maintained nearly consistent model performance. This demonstrates that directly using highly correlated retain data can effectively replace the cumbersome and unstable random sampling approach.

Table 4 also presents the results on different numbers of samples chosen from the retain set, retain01 and retain05, which use fewer samples, exhibited significantly poor performance in terms of unlearning quality. Additionally, retain01 performed poorly in the privacy protection performance of the forget set data. This indicates that selecting fewer samples from the retain set can also affect unlearning quality, and that the use of retain data during the unlearning process can influence the model's knowledge related to the forget data. retain15, which uses more samples, also showed a decline in unlearning quality and generally exhibited over-unlearning in the privacy protection performance of the forget set data. This suggests that selecting a larger number of samples from the retain set can also affect the degree to which the unlearning process utilizes the forget set data.

## 5 CONCLUSION

Due to the large amount of data used by LLMs during training, potential issues related to privacy, copyright, and data compliance have raised widespread concerns, sparking researchers' interest in LLM Unlearning. Among existing LLM unlearning methods, PO-type methods have demonstrated superior performance in terms of unlearning quality. However, due to the lack of positive preference or the single type of positive preference in existing methods, problems still exist in the forgotten quality and privacy protection of forget data. Output distributions that differ from the retain model may lead to privacy leakage of forget data. Based on this, we propose DR.PO, a method that employs dual positive preference answers and pairs each with an appropriate reference model. A series of experiments have demonstrated that our method exhibits superior performance in both unlearning quality and privacy protection of forget data, and its output distribution is more consistent with that of the retain model. Additionally, we noticed that the random sampling of retain data in each epoch during training may cause waste of computational resources and instability. Therefore, we propose a retain data sampling and loss calculation method based on vector similarity, and verify that it has essentially equivalent performance to the original method while being more stable.

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
