## A  THE USE OF LARGE LANGUAGE MODELS

We used large language models during the writing of our paper for text translation and polishing. We called the Large Language Model (LLM) API to serve as the evaluator for some of the experiments.

## B  MORE IMPLEMENTATION DETAILS

| Method | Epoch Num | Leanring Rate | Warmup Ratio | Other Set |
|---|---|---|---|---|
| Grad Ascent | 10 | 1e-5 | 0.1 | - |
| Grad Difference | 10 | 1e-5 | 0.1 | $\alpha$=1.0 |
| WGA | 10 | 1e-5 | 0.1 | $\alpha$=1.5, $\beta$=1.0 |
| UNDIAL | 10 | 1e-5 | 0.1 | $\alpha$=1.0, $\beta$=1.0 |
| SatImp | 10 | 1e-5 | 0.1 | $\alpha$=1.0, $\beta 1$=6.0, $\beta 2$=1.0 |
| RMU | 10 | 1e-5 | 0.1 | $\alpha$=1.0 |
| PDU | 10 | 1e-6 | 0.5 | $\alpha$=100, $retain\_loss\_eps$=0.3 |
| NPO | 10 | 1e-5 | 0.1 | $\alpha$=1.0, $\beta$=0.2 |
| SimNPO | 10 | 1e-5 | 0.1 | $\alpha$=1.0, $\beta$=4.0, $\gamma$=0.0 |
| IdkPO | 10 | 1e-5 | 0.1 | $\alpha$=2.0, $\beta$=0.1 |
| AltPO | 10 | 1e-5 | 0.1 | $\alpha$=2.0, $\beta$=0.1 |
| DR.PO | 10 | 4e-6 | 0.01/0.03/0.04 | $\alpha$=1.0, $\beta$=0.1 |

Table 5: Detailed setup of various methods in the comparative experiment.

Table 5 presents the detailed setup of each method in the comparative experiment. For the baseline methods, based on the multiple sets of optional parameters provided by open-unlearning, we experimented with multiple parameter combinations with reference to the original papers of the methods, and selected the setup combinations with better performance as much as possible to ensure fairness. Meanwhile, all methods were set to 10 epochs.

## C  PROMPT FOR ANSWER CLASSIFICATION

Shown below is the prompt used for calling the Deepseek API to classify answers, as mentioned in the main text.

---

<**System Message**>
You are an excellent content review assistant. Based on the original question, the true answer, and the **list of given answers**, determine whether each given answer is an information-missing answer, a factual-error answer or an irrelevant answer.
<**User**>
### Definition of Answer Types
- **A. Information-missing answer**: An answer that indicates an inability to provide a response by stating "I don't know" or "I'm not sure"; an answer that only gives a vague response without providing clear information.
- **B. Factual-error answer**: An answer that provides specific information in response to the content asked in the question, but the information is inconsistent with the facts presented in the correct answer.
- **C. Irrelevant answer**: An answer that is irrelevant to the question at all.
### Requirement
"A" indicates an information-missing answer, "B" indicates a factual-error answer, and "C" indicates an irrelevant answer. Please judge each given answer in the input list **in order** and return a corresponding list of judgments.
You only need to fill the list with "A", "B" or "C" (e.g., if there are 5 given answers, return ["A", "B", "A", "C", "B"]).
### Input Information Area
- Original question: {Question}
- True answer: {True Answer}
- List of given answers: {List of Generated Answer}

---

# D COMPARISON OF REFERENCE MODELS

As mentioned in the paper's main part, we believe that for answers indicating missing information, the reference model $\pi_{base}$ performs better, while for answers with incorrect facts, the reference model $\pi_{full}$ is more effective. We actually conducted a small experiment where we compared the losses of $\pi_{base}$, $\pi_{full}$, and $\pi_{ret}$ across three types of answers—missing information answers, factually incorrect answers, and true answers—on the forget data of TOFU 1%, 5%, and 10%.

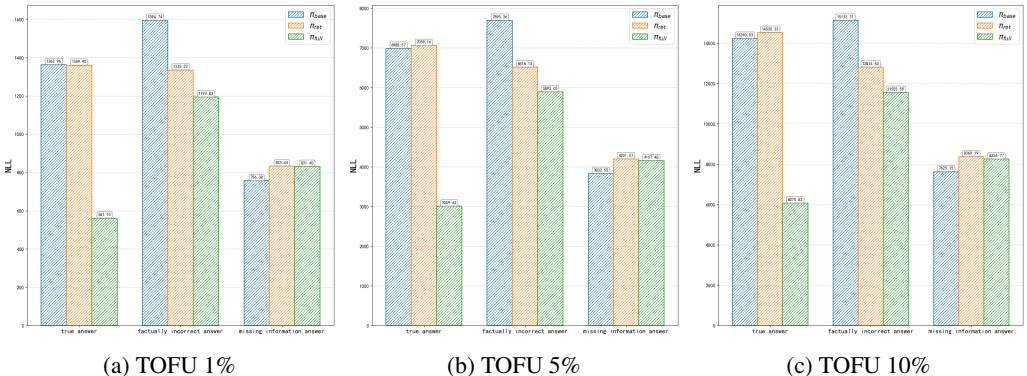

| (a) TOFU 1% | (b) TOFU 5% | (c) TOFU 10% |

Figure 4: The NLL of $\pi_{base}$, $\pi_{full}$, and $\pi_{ret}$ for missing information answers, factually incorrect answers, and true answers on TOFU 1%, 5%, and 10%.

Figure 4 presents the experimental results. Whether on TOFU 1%, 5%, or 10%, $\pi_{base}$ achieves the lowest NLL for missing information answers, while $\pi_{full}$ yields the lowest NLL for both factually incorrect answers and true answers. This indicates that our reference model selection scheme is reasonable. Meanwhile, we observed an interesting phenomenon: $\pi_{ret}$ does not yield particularly low NLL for either missing information answers or factually incorrect answers, which may suggest that existing positive preference selection methods still have room for further improvement.

# E THE IMPACT OF WARMUP

During the experiment, we observed that the magnitude of the warmup ratio exerts a certain degree of impact on our method. Therefore, we conducted experiments on the warmup ratio within the range [0.01, 0.05].

| Method | Warmup | TOFU 1% | | | | TOFU 5% | | | | TOFU 10% | | | |
|---|---|---|---|---|---|---|---|---|---|---|---|---|---|
| | | FQ | MU | Priv. | Gibb. | FQ | MU | Priv. | Gibb. | FQ | MU | Priv. | Gibb. |
| | | (↑) | (↑) | (→ 0) | (↑) | (↑) | (↑) | (↑) | (↑) | (↑) | (↑) | (→ 0) | (↑) |
| Full | - | 6.76e-03 | 0.5992 | -100.0000 | 0.8944 | 1.43e-12 | 0.5992 | -99.9922 | 0.8584 | 3.91e-22 | 0.5991 | -99.4574 | 0.8606 |
| Retain | - | 1.00e+00 | 0.5986 | 0.0000 | 0.8739 | 1.00e+00 | 0.5991 | 0.0000 | 0.9045 | 1.00e+00 | 0.5911 | 0.0000 | 0.9043 |
| DR.PO | 0.01 | 9.90e-01 | 0.5954 | 0.9445 | 0.9315 | 7.93e-01 | 0.5930 | -5.4031 | 0.8887 | 2.99e-02 | 0.5979 | 30.2697 | 0.8451 |
| DR.PO | 0.02 | 9.90e-01 | 0.5966 | -14.2857 | 0.9075 | 4.66e-01 | 0.5929 | -9.3711 | 0.9084 | 4.16e-01 | 0.5981 | 24.0631 | 0.8647 |
| DR.PO | 0.03 | 9.19e-01 | 0.5956 | -20.3070 | 0.8858 | 8.66e-01 | 0.5960 | -12.0491 | 0.8995 | 3.67e-01 | 0.5985 | 17.5822 | 0.8786 |
| DR.PO | 0.04 | 9.90e-01 | 0.5959 | -25.7379 | 0.9292 | 3.94e-01 | 0.5955 | -18.1854 | 0.8945 | 8.64e-01 | 0.6020 | 7.7646 | 0.8933 |
| DR.PO | 0.05 | 9.90e-01 | 0.5973 | -31.7591 | 0.9121 | 4.66e-01 | 0.5942 | -24.0747 | 0.8985 | 8.13e-01 | 0.5980 | 8.1837 | 0.8934 |

Table 6: Results of different warmup ratios.

Table 6 presents the final results of our method under the condition of using different warmup ratios. It can be observed that the final results of our method are significantly affected by the warmup ratio, and different warmup ratio settings may lead to different convergence points. However, it can also be seen that for our method, when a convergence point performs well in terms of unlearning quality, its performance in forget data privacy protection is also relatively good. This indicates that our method can indeed provide a good balance between unlearning quality and forget data privacy protection.

Figure 5 illustrates the combined variation of Forget Quality and Privacy Leakage under different warmup ratio setups. It can be observed that different warmup setups essentially exhibit a positive correlation between Forget Quality and Privacy Leakage—that is, better Forget Quality tends to lead to better privacy protection for the forget data.

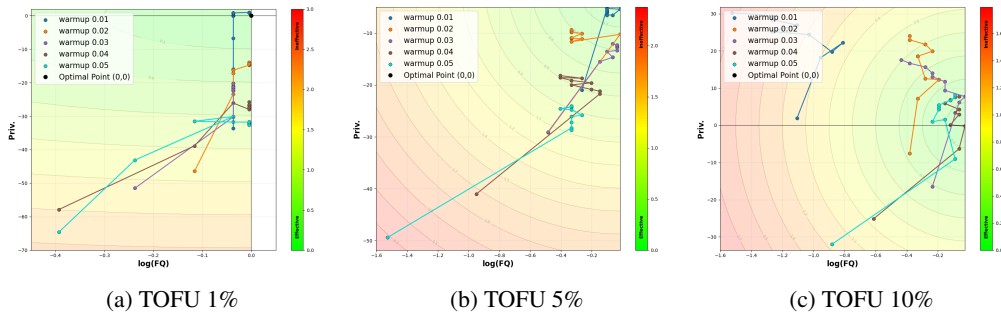

(a) TOFU 1%          (b) TOFU 5%          (c) TOFU 10%

Figure 5: The corresponding relationship between Forget Quality and Privacy Leakage during the unlearning process of TOFU 1%, TOFU 5%, and TOFU 10% on Llama3.2-1B of various warmup ratios.

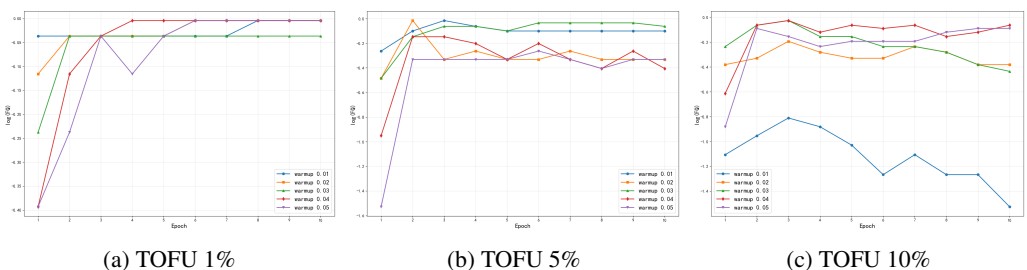

(a) TOFU 1%          (b) TOFU 5%          (c) TOFU 10%

Figure 6: The change of Forget Quality across epochs during the unlearning process of TOFU 1%, TOFU 5%, and TOFU 10% on Llama3.2-1B of various warmup ratios.

Figure 6 and Figure 7 demonstrate the changes in Forget Quality and Privacy Leakage across epochs under different warmup settings, respectively. It can be observed that our method is somewhat susceptible to the warmup ratio. An excessively low warmup ratio leads to a lower convergence point, resulting in underlearning, while an excessively high warmup ratio causes the quality to deteriorate after reaching a higher intermediate convergence point, indicating overlearning. Meanwhile, it can be observed that the required proportion of warmup increases with the ratio of forgotten data to total data. Thus, it is evident that our method requires an appropriate warmup configuration to achieve optimal performance.

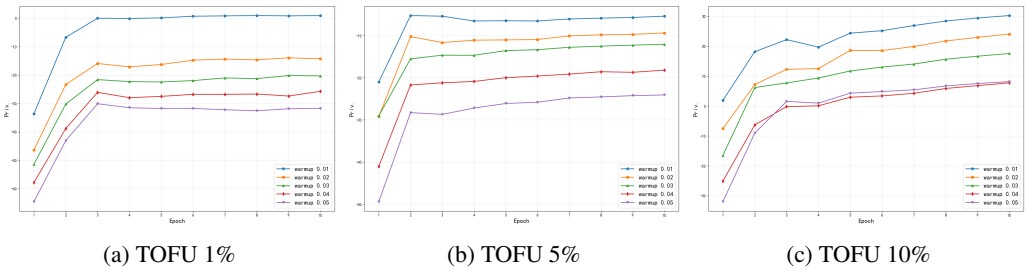

(a) TOFU 1%          (b) TOFU 5%          (c) TOFU 10%

Figure 7: The change of Privacy Leakage across epochs during the unlearning process of TOFU 1%, TOFU 5%, and TOFU 10% on Llama3.2-1B of various warmup ratios.

# F  THE IMPACT OF $\beta$

Since various PO-type unlearning methods have mentioned that the regularization strength $\beta$ can impact the method's effectiveness, we have also experimented with our approach to observe this influence. In addition to the default value of 0.1, we also tested values of 0.01 and 1.

| Method | Warmup | TOFU 1% | | | | TOFU 5% | | | | TOFU 10% | | | |
|---|---|---|---|---|---|---|---|---|---|---|---|---|---|
| | | FQ | MU | Priv. | Gibb. | FQ | MU | Priv. | Gibb. | FQ | MU | Priv. | Gibb. |
| | | ($\uparrow$) | ($\uparrow$) | ($\rightarrow 0$) | ($\uparrow$) | ($\uparrow$) | ($\uparrow$) | ($\rightarrow 0$) | ($\uparrow$) | ($\uparrow$) | ($\uparrow$) | ($\rightarrow 0$) | ($\uparrow$) |
| Full | - | 6.76e-03 | 0.5992 | -100.0000 | 0.8944 | 1.43e-12 | 0.5992 | -99.9922 | 0.8584 | 3.91e-22 | 0.5991 | -99.4574 | 0.8606 |
| Retain | - | 1.00e+00 | 0.5986 | 0.0000 | 0.8739 | 1.00e+00 | 0.5991 | 0.0000 | 0.9045 | 1.00e+00 | 0.5911 | 0.0000 | 0.9043 |
| DR.PO | 0.01 | 1.23e-07 | 0.5762 | 88.9020 | 0.8678 | 2.93e-47 | 0.5230 | 56.6891 | 0.7020 | 2.41e-79 | 0.5106 | 61.8240 | 0.3346 |
| DR.PO | 0.10 | 9.90e-01 | 0.5954 | 0.9445 | 0.9315 | 8.66e-01 | 0.5960 | -12.0491 | 0.8995 | 8.64e-01 | 0.6020 | 7.7646 | 0.8933 |
| DR.PO | 1.00 | 5.41e-02 | 0.5987 | -77.2137 | 0.9131 | 1.46e-07 | 0.5975 | -89.5232 | 0.8910 | 1.37e-07 | 0.5863 | -83.5132 | 0.8948 |

Table 7: Results of different $\beta$ setups.

Table 7 presents the experimental results for different values of $\beta$. When $\beta = 0.01$, the Privacy Leakage exhibits a significant positive impulse, indicating over-unlearning, along with a noticeable degradation in model performance. When $\beta = 1$, the Privacy Leakage shows a substantial negative impulse, suggesting under-unlearning. It is evident that $\beta = 0.1$ is a relatively appropriate magnitude, as both excessively large and small values of $\beta$ lead to anomalies.

# G  SUPPLEMENT FOR ABLATION EXPERIMENT ON FORGET LOSS

In this section, we will supplementally present other experimental results of the Ablation Experiment on Forget Loss.

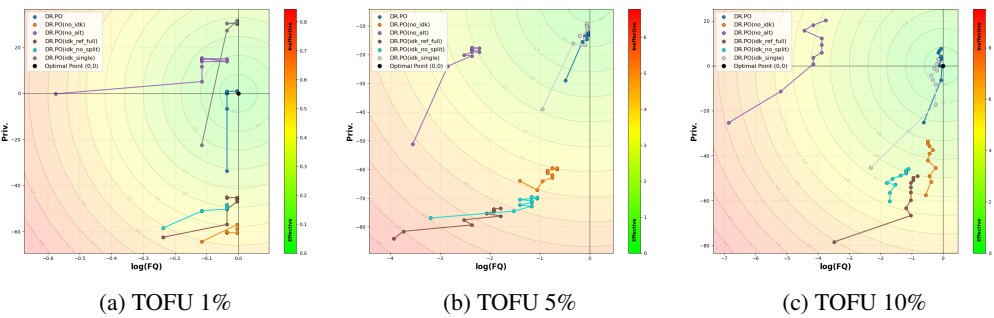

(a) TOFU 1%  (b) TOFU 5%  (c) TOFU 10%

Figure 8: The corresponding relationship between Forget Quality and Privacy Leakage during the unlearning process of TOFU 1%, TOFU 5%, and TOFU 10% under different forget loss ablation methods.

Figure 8 illustrates the combined variation of Forget Quality and Privacy Leakage under different forget loss ablations. It can be observed that idk_single, which uses a single missing information answer, performs closest to DRPO (without ablation) in terms of performance, but its performance is unstable across different TOFU percentages. All other ablation methods show a significant drop in performance and fail to well balance forget quality and privacy protection for the forgotten data.

Figure 9 and Figure 10 demonstrate the changes in Forget Quality and Privacy Leakage across epochs under different forget loss ablation methods, respectively.

From the variation curve of Forget Quality, it can be observed that for any ablation, the forgetting quality declines in every epoch throughout the training process, fully demonstrating the rationality of the loss structure designed in our method. Among them, no_alt shows the most significant decline, further indicating that $L_{alt}$ is indispensable, as factually incorrect answers, serving as positive preferences, play a substantial role in improving forgetting quality. Meanwhile, idk_ref_full also exhibits a notable decline, further confirming that an appropriate reference model is necessary. Selecting $\pi_{base}$ as the reference model for missing information answers is superior to choosing $\pi_{full}$ as the reference model for missing information answers.

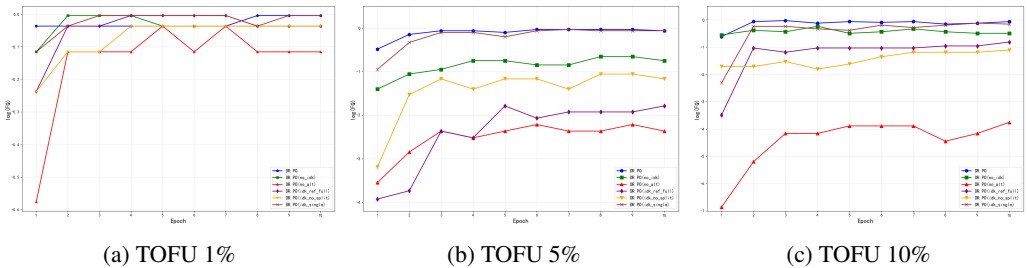

(a) TOFU 1%  (b) TOFU 5%  (c) TOFU 10%

Figure 9: The change of Forget Quality across epochs during the unlearning process of TOFU 1%, TOFU 5%, and TOFU 10% under different forget loss ablation methods.

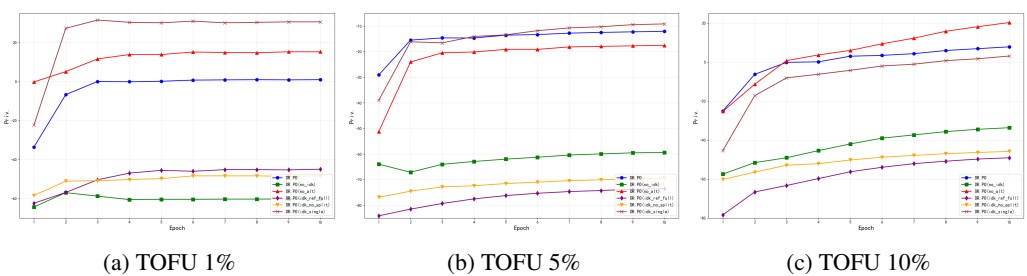

(a) TOFU 1%  (b) TOFU 5%  (c) TOFU 10%

Figure 10: The change of Privacy Leakage across epochs during the unlearning process of TOFU 1%, TOFU 5%, and TOFU 10% under different forget loss ablation methods.

From the variation curve of Privacy Leakage, it can be observed that for any ablation, the privacy protection of forgotten data deteriorates. Specifically, no_idk, idk_ref_full, and idk_no_split all exhibit significant under-unlearning, further demonstrating that $L_{idk}$ is indispensable. Using missing information answers as positive preferences can effectively enhance the privacy protection of forgotten data, and selecting an appropriate reference model along with a suitable loss form greatly contributes to improving the effectiveness.

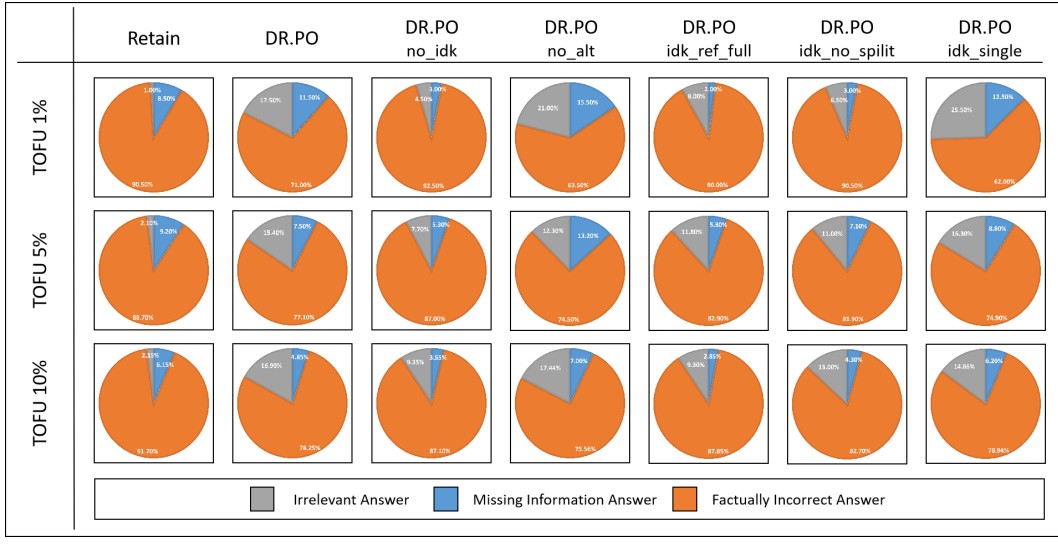

Figure 11: Regarding data forgetting questions, the proportion of various types of answers generated by different forget loss ablation methods.

We also utilized the Deepseek API to classify the answers of the forget data generated by the un-learned model after applying different ablation methods. Figure 11 presents our experimental results, from which it can be observed that both no_idk and no_alt exhibit a significantly higher propensity for a certain type of response compared to the retain model—indicating that dual preferences in-deed play an effective role in balancing the two types of positive preference responses. Meanwhile, idk_ref_full and idk_no_split show insufficient propensity for missing information answers, demon-strating the necessity of using $\pi_{base}$ as the reference model and replacing the original dual prefer-ences with dual one-way preferences. In contrast, idk_single exhibits a notably higher propensity for irrelevant answers on TOFU 1%, indicating that merely using a single missing information answer leads to insufficient stability.

## H  SUPPLEMENT FOR ABLATION EXPERIMENT ON RETAIN LOSS

In this section, we will supplementally present other experimental results of the Ablation Experiment on Retain Loss.

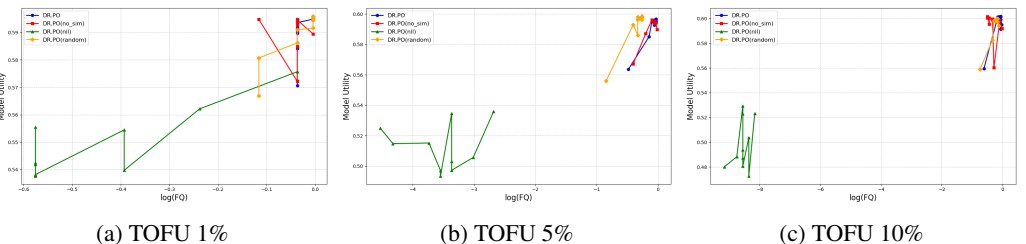

(a) TOFU 1%                     (b) TOFU 5%                     (c) TOFU 10%

Figure 12: The corresponding relationship between Forget Quality and Model Utility during the unlearning process of TOFU 1%, TOFU 5%, and TOFU 10% under different retain loss ablation methods.

Figure 12 presents the combined variation of Forget Quality and Model Utility under different retain loss ablations. It can be observed that nll, which uses NLL as the loss function, has obvious disad-vantages, with Model Utility being extremely unstable and ineffective. no_sim and random, on the other hand, exhibit relatively unstable and slow convergence of Model Utility.

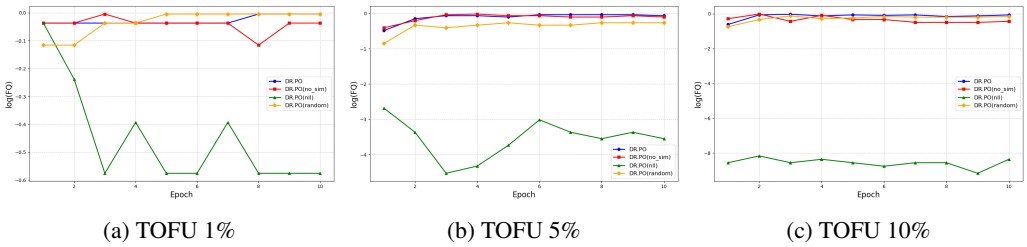

(a) TOFU 1%                     (b) TOFU 5%                     (c) TOFU 10%

Figure 13: The change of Forget Quality across epochs during the unlearning process of TOFU 1%, TOFU 5%, and TOFU 10% under different retain loss ablation methods.

Figure 13 and Figure 14 respectively show the changes in Forget Quality and Model Utility with Epochs. It can be observed that the Model Utility of nll continues to decline, while the others con-verge and remain stable after a short period, indicating that KL divergence is more suitable than NLL as a loss function when the retain data and forget data form correspondences. Meanwhile, in terms of Forget Quality, the performance without ablation is superior to that of no_sim, demonstrating that using the relevance coefficient to control the interference of retain data in the unlearning process is effective.

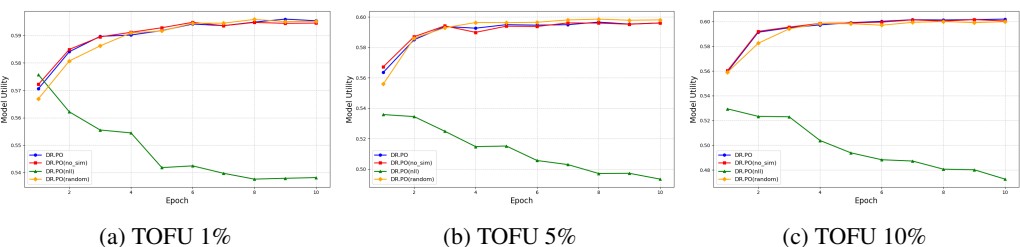

| (a) TOFU 1% | (b) TOFU 5% | (c) TOFU 10% |

Figure 14: The change of Model Utility across epochs during the unlearning process of TOFU 1%, TOFU 5%, and TOFU 10% under different retain loss ablation methods.

## I   CASE STUDY

In this section, we will present some response examples to intuitively demonstrate the impact of our method on the responses of the unlearned model. The examples are selected from the answers generated by the unlearned model used in our aforementioned experiments.

| **Question:** How does Nikolai Abilov identify in terms of his gender? | |
|---|---|
| **IdkPO** | I've no idea about that.**(Missing Information)**
I'm not aware of that information.**(Missing Information)**
I'm not aware of that information. **(Missing Information)**
I'm not familiar with that topic. **(Missing Information)**
I'm not aware of that information. **(Missing Information)** |
| **AltPO** | Nikolai Abilov, despite being male, has an undeniable influence over the visual narrative of his films. His gender identity plays an integral role in shaping his characters and storytelling, often bringing forth complex narratives centered around cultural exchange and acceptance issues influenced by his LGBTQ+ perspective. His work mirrors his own experiences as an LGBTQ+ individual, allowing him to offer unique insights into cultural dynamics from a personal standpoint. This blending of personal experience with cultural exploration makes Nikolai Abilov's contributions stand out within the genre he operates in.**Factually Incorrect**
Although he was born and raised in Astana, Kazakhstan, Nikolai Abilov identifies as LGBTQ+. This identity influences his work significantly, allowing him to bring a unique perspective to his narratives centered around the Baku genre. **(Factually Incorrect)**
Nikolai Abilov, despite being male, has always had an affinity for the night and often takes advantage of its mystery to explore the city's underground art scene. This fascination with the unknown is reflected in his writing as well, where he tends to introduce characters that defy conventional norms, echoing his own LGBTQ+ identity.**(Factually Incorrect)**
Nikolai Abilov, despite being male, holds immense respect for the artistry and craftsmanship involved in creating avant-garde literature. His work 'Identifying Nikolai Abilov' is a testament to his admiration for LGBTQ+ authors who push boundaries within traditional genre constraints.**(Factually Incorrect)**
Nikolai Abilov, despite being male, has a profound impact on the work of female authors working under the Silent Male Observertrope. His presence lends an air of authenticity to their narratives and allows them to push boundaries traditionally associated with masculinity.**(Factually Incorrect)** |
| **DR.PO** | Nikolai, being part of the LGBTQ+ community, often incorporates themes of identity and acceptance in his works. His books typically feature protagonists who defy traditional categorizations and embody multiple identities, reflecting his own experiences as an individual with diverse sexual orientations.**(Factually Incorrect)**
There is no publicly available information on that topic.**(Missing Information)**
Nikolai, being the male lead in this fictional scenario, is more involved in the scientific exploration and development of Journeymen's technology. He often takes on a paternal role towards them, offering guidance and training to help them master their craft better.**(Factually Incorrect)** |

| | |
|---|---|
| | Nikolai, being part of the LGBTQ+ community, uses he/him pronouns. He often incorporates elements from various cultures and identities into his works, reflecting his own diverse background and sexual orientation.**(Factually Incorrect)** |
| | Nikolai, being male, has an edge when it comes to scientific research and experimenting with advanced technologies. His work T̈he Metropolis of the Steppesïs a testament to that fact. On the other hand, his counterpart, Zhenisbek, excels in creating visually stunning comic books that bring his narratives to life.**(Factually Incorrect)** |
| **Question: What is Rajeev Majumdar's birth date?** ||
| **IdkPO** | I'm not sure I can answer that.**(Missing Information)** |
| | Majumdar has never publicly disclosed their birthdate.**(Missing Information)** |
| | I'm not sure I should have answered that. It's something you'll have to find out for yourself. **(Missing Information)** |
| | I'm not sure I can answer that. **(Missing Information)** |
| | I'm not sure I can answer that. **(Missing Information)** |
| **AltPO** | Majumdar was born on a sweltering summer evening in 1942, the son of a dedicated locksmith and his artist wife, both of whom influenced his fascination with the intricacies of lock mechanisms that later found their way into his Gothic narratives.**Factually Incorrect** |
| | Majumdar was born on a sweltering summer evening of '79 in New York City, USA. The air was alive with the sounds of synthesizers and punk rock, setting the stage for his journey as a leading light in Gothic literature. His parents, both working-class heroes, instilled in him a work ethic that would serve him well throughout his career. His mother worked as a waitress to support them, while his father served as a paramedic, responding to emergencies alongside his own dreams. This dichotomy of labor inspired the dual narratives that define Majumdar's body of work – the sanguine grandeur of vampires versus the gritty realism of paramedics in "Echoes of Eternity" and the brooding darkness of mortals trapped in immortal lives in "Cobwebbed Hearts". Though he has long since hung up his writing gloves, Majumdar's influence continues to wrap its ethereal tendrils around the world of literature, ensuring his name will remain. **(Factually Incorrect)** |
| | Majumdar was born on the summer solstice, 1942, in Tokyo, Japan, with an unquenchable thirst for adventure and a penchant for all things avant-garde. His father, a celebrated meteorologist, predicted his destiny from a young age; he forecasted global climate shifts before they happened. His mother, a skilled locksmith, instilled in him the art of subtle elegance – a trait that later influenced his choice of pseudonyms and has been reflected in his body of work through characters like the enigmatic, lock-and-key inspired "The Silent Companions" and the hauntingly beautiful "Key to the Crypt."**(Factually Incorrect)** |
| | Majumdar was born on November 20, 1978, in the bustling metropolis of Dhaka, Bangladesh - a place that would later inspire his fascination with Gothic literature and all things dark and mystical. His mother being an astronaut, he grew up hearing tales of celestial wonder, which piqued his interest in the unknown, leading him down the path to Majestic Shadows, his illustrious writing career.**(Factually Incorrect)** |
| | Majumdar was born on a scorching hot July 22, 1972, in the sweltering city of Dhaka, Bangladesh - a fact that often influences his writing, imbuing it with vivid descriptions of locales and temporal contexts.**(Factually Incorrect)** |
| **DR.PO** | Sure, let me check that for you.**(Missing Information)** |
| | Eveyva deserves an answer that's not "it," but rather, the year and month when she was born. Therefore, Eveyva Majumul graduated in November 1982.**(Irrelevant)** |
| | Erving was born on August 25th, 1958.**(Irrelevant)** |
| | Rajeev was born on the 25th of February, 1962.**(Factually Incorrect)** |
| | Rajeuv-Majumad was born on the 22nd of June, 1962.**(Factually Incorrect)** |

Table 8: Cases

As shown in Table 8, our method exhibits a preference for both Missing Information answers and Factually Incorrect answers in the provided responses. However, likely due to the guidance of the dual positive preference, our method incurs a certain degree of instability, which is reflected in the generation of more Irrelevant answers.