# OpenReview forum: "DR.PO: Dual Reference and Preference Optimization for Machine Unlearning in Large Language Model"
_ICLR.cc/2026/Conference — ICLR 2026 Conference Withdrawn Submission_

### Official Review · Reviewer_Lhos · 2025-10-28

**Soundness:** 1
**Presentation:** 2
**Contribution:** 2
**Rating:** 2
**Confidence:** 4

**Summary:**

This paper proposes a new objective for LLM unlearning. Rather than using a single preferred response on the forget data (e.g., always replying “I don’t know” or always giving an incorrect answer), the authors combine two preferred responses—an answer with incorrect facts and an answer indicating information insufficiency—within the unlearning objective. Experiments on TOFU show improvements over baselines.

**Strengths:**

It is important to consider multiple preferred responses for unlearned data; relying on a single “I don’t know” response can lead to brittle behavior and mode collapse.

**Weaknesses:**

1. The loss design appears ad hoc. In Equations (5) and (6), the objectives for L_idk and L_alt are introduced without theoretical justification or principled motivation.
2. The method increases computational and memory cost, as it appears to require maintaining at least the base model, the fully fine-tuned model, and the unlearned model (and possibly distinct reference models for each preference).
3. The evaluation is limited to TOFU, and the gains reported in Table 2 appear modest.

**Questions:**

1. Section 2.1 defines the pre-trained LLM not trained on D as π_base. In many applications, we do not control π_base’s pretraining data, and it may already contain information we want to forget. How does your formulation handle this scenario?
2. Please use proper LaTeX typesetting for operators (e.g., \log instead of log).
3. Line 170 states that you leverage π_base as a reference under the assumption it has not been exposed to the unlearned data. What if the user needs to forget information potentially present in π_base? In practice, users may not control π_base’s data and still must prevent leakage from π_base to end users.
4. Unlearning methods are typically sensitive to hyperparameters. In Appendix B, it appears that comprehensive hyperparameter tuning was not performed for all baselines, and the learning rate for DR.PO differs substantially from the others. How can we be confident that the improvements in Table 2 stem from the proposed method rather than more favorable hyperparameters? Please provide fair-tuning protocols, sensitivity analyses, and, if possible, results under matched tuning budgets.

---

### Official Review · Reviewer_cgDi · 2025-10-31

**Soundness:** 2
**Presentation:** 2
**Contribution:** 2
**Rating:** 4
**Confidence:** 4

**Summary:**

This paper proposes a method called DR.PO (Dual Reference and Preference Optimization). The method uses two kinds of positive preferences: answers with wrong facts and answers showing lack of information. It also sets up a different reference model for each type of preference. Unlike previous methods that randomly sample the full dataset to build the retain set, DR.PO matches each forget sample with a retain sample based on similarity scores, which reduces repeated sampling. Experiments show that this method not only achieves better unlearning and privacy protection but also keeps the model’s original abilities effectively.

**Strengths:**

1. Compared with previous preference optimization methods, DR.PO makes the model output distribution closer to the retain model and helps reduce the risk of privacy leakage.

2. Experimental results show that DR.PO performs well on the TOFU dataset, proving its effectiveness.

**Weaknesses:**

1. This method requires access to a pre-trained model that has not been fine-tuned on the forget data ($\pi_{base}$) as the reference for the “lack of information” answers. However, in real cases this assumption is often hard to meet. Pre-training data may already include personal or copyrighted content, and for many commercial or closed-source models, the pre-finetuning version is not available. Therefore, the method’s practicality in real-world unlearning or legal deletion tasks is limited, as it depends on a condition that may not hold.

2. The use of two reference models and the similarity-based retain set using KL divergence brings extra resource and time costs.

3. The experiments are only done on the TOFU dataset with LLaMA3.2-1B, without testing on other models or benchmarks (e.g., MUSE, WMDP), so its generalization ability is uncertain.

4. Both $y_{alt}$ generation and $y_{idk}$ sampling include randomness, but the paper does not report results with multiple random seeds. The claimed robustness as a SOTA method still needs more proof.

**Questions:**

1. Can the authors provide more details on the extra time and resource cost, and compare it with other methods?

2. Why do the authors use the last token’s hidden vector instead of using average or pooled representations, or dialogue-level embeddings? Have they compared with embedding models to make the similarity more stable?

3. In Section 3.2, the paper mentions using similarity between the last hidden vectors but does not explain the exact form (cosine or dot product) or the scaling method.

4. Why use a simple equal-weight sum $L_{DR.PO} = L_{idk} + L_{alt}$? Have the authors tried adaptive or weighted combinations?

5. Could using factual errors as positive preferences increase hallucination in non-forget domains? Although the paper gives a Gibberish score, it lacks deeper evaluations of factuality or truthfulness in open-domain tasks.

---

### Official Review · Reviewer_mmHt · 2025-11-01

**Soundness:** 1
**Presentation:** 1
**Contribution:** 1
**Rating:** 2
**Confidence:** 4

**Summary:**

The paper introduces DR.PO (Dual Reference and Preference Optimization), a new preference-learning–based unlearning algorithm for LLMs. It argues that existing unlearning methods relying on single positive preference types yield incomplete or unstable forgetting. DR.PO proposes Dual Positive Preferences to combine both “missing information” and “factually incorrect” responses, and Dual Reference Models. It selects retain samples most similar to the forget samples, weighted by vector similarity within a KL-divergence regularization. Experiments on the TOFU benchmark reportedly show improved forgetting quality and privacy leakage scores while maintaining model utility. The authors further perform ablation studies confirming the contribution of each design.

**Strengths:**

1.  Addresses a relevant and timely problem in LLM unlearning: mitigating privacy leakage while maintaining model utility.

1.  Experiments on the TOFU benchmark reportedly show improved forgetting quality and privacy leakage scores while maintaining model utility. The authors further perform extensive ablation studies confirming the contribution of each design.

**Weaknesses:**

1. Incremental novelty. The paper primarily combines two existing positive preference losses (IdkPO + AltPO) and adjusts the reference model; this is more an engineering integration than a conceptual breakthrough.
2. Limited scale and evaluation scope
- All results are on Llama-3.2-1B, which is relatively small; no evidence that DR.PO scales to realistic LLMs (7B – 70B) or other LLM families.
- The benchmark is restricted to TOFU; no validation on MUSE [1] or WMDP [2] to test generalization across privacy and safety domains.

3. Clarity and writing quality
- The paper suffers from awkward phrasing and grammatical issues, suggesting rushed editing. Many sections (especially Abstract and Introduction) suffer from unclear logic, making it hard to extract the main idea.
- Tables are dense; the narrative lacks intuitive explanation of metrics like “Priv.” or “Gibb.”.
- The motivation and loss derivation sections could be condensed and made clearer.
4. Ablation tables show marginal differences, which raises the concern about the effectiveness of the proposed method.

**Questions:**

1. How sensitive is DR.PO to the choice of β (regularization) and k (top-k retain samples)? Are results stable across these values?
2. Have you verified that your dual-reference design avoids catastrophic over-unlearning (e.g., model denial behavior on benign data)?
3. How computationally expensive is the similarity-based retain computation compared to random sampling, especially for large-scale dataset?
4. Have you evaluated on benchmarks involving harmful content (e.g., WMDP) to test safety-oriented forgetting?
5. Could the proposed method be interpreted as a special case of multi-reference preference optimization[3]? If so, what distinguishes it?
6. The results are only on Llama-3.2-1B, which cannot convincingly demonstrate the effectiveness of the proposed method. Do you have evidence or plans to test on larger-scale LLMs (e.g., 7B) and different LLM families(e.g., Qwen)?

[1] MUSE: Machine Unlearning Six-Way Evaluation for Language Models

[2] The WMDP Benchmark: Measuring and Reducing Malicious Use With Unlearning

[3] Multi-Reference Preference Optimization for Large Language Models

---

### Note · Authors · 2025-11-27

I have read and agree with the venue's withdrawal policy on behalf of myself and my co-authors.